Being there: a brief visit to a neighbourhood induces the social attitudes of that neighbourhood

Nettle Daniel 1 daniel.nettle@ncl.ac.uk
Pepper Gillian V. 1
Jobling Ruth 1
Schroeder Kari Britt 1 2
1 Centre for Behaviour & Evolution, Newcastle University , Newcastle , UK
2 Department of Psychology, Boston University , Boston, MA , USA
Cloninger C. Robert
Electronic publication date: 2014 Jan 14
Publication date: 2014
Volume: 2
Electronic Location ID: e236
Received 2013 Oct 4; Accepted 2013 Dec 11
Copyright: © 2014 Nettle et al.
Copyright year: 2014
Copyright holder: Nettle et al.
License: This is an open access article distributed under the terms of the Creative Commons Attribution License, which permits unrestricted use, distribution, and reproduction in any medium, provided the original author and source are credited.
License URL: https://creativecommons.org/licenses/by/3.0/

Keywords: Neighbourhood effects, Paranoia, Trust, Cultural evolution, Social disorder, Mental health, Social capital

Funding: National Science Foundation Award 1003961 (Kari Britt Schroeder) This work was funded by the US National Science Foundation, the Graduate School of the Faculty of Medical Sciences, Newcastle University, and the School of Psychology, Newcastle University. The funders had no role in study design, data collection and analysis, decision to publish, or preparation of the manuscript.

==============================
There are differences between human groups in social behaviours and the attitudes that underlie them, such as trust. However, the psychological mechanisms that produce and reproduce this variation are not well understood. In particular, it is not clear whether assimilation to the social culture of a group requires lengthy socialization within that group, or can be more rapidly and reversibly evoked by exposure to the group’s environment and the behaviour of its members. Here, we report the results of a two-part study in two neighbourhoods of a British city, one economically deprived with relatively high crime, and the other affluent and lower in crime. In the first part of the study, we surveyed residents and found that the residents of the deprived neighbourhood had lower levels of social trust and higher levels of paranoia than the residents of the affluent neighbourhood. In the second part, we experimentally transported student volunteers who resided in neither neighbourhood to one or the other, and had them walk around delivering questionnaires to houses. We surveyed their trust and paranoia, and found significant differences according to which neighbourhood they had been sent to. The differences in the visitors mirrored the differences seen in the residents, with visitors to the deprived neighbourhood reporting lower social trust and higher paranoia than visitors to the affluent one. The magnitudes of the neighbourhood differences in the visitors, who only spent up to 45 min in the locations, were nearly as great as the magnitudes of those amongst the residents. We discuss the relevance of our findings to differential psychology, neighbourhood effects on social outcomes, and models of cultural evolution.

Introduction

There are substantial differences between human groups in social behaviours and the attitudes that underlie them. Much of the literature demonstrating these differences has compared different ethnic or national groups (e.g., Gachter & Herrmann, 2009; Henrich et al., 2005; Henrich et al., 2010; Herrmann, Thoni & Gachter, 2008). However, differences at a much smaller scale, such as villages within one ethnic population or neighbourhoods within one city, can be equally marked (Falk & Zehnder, 2007; Gurven, Zanolini & Schniter, 2008; Lamba & Mace, 2011; Nettle, Colléony & Cockerill, 2011; Wilson, O’Brien & Sesma, 2009). Whilst these observations are relatively novel, they are conceptually related to what can broadly be termed neighbourhood effects, which have been intensely studied in social science for several decades. The literature on neighbourhood effects is concerned with the consequences of the features of the immediately surrounding ecology for outcomes such as criminality, violent conduct, antisocial behaviour, trust, paranoia, and depression, which are clearly related to social behaviour (see Aneshensel & Sucoff, 1996; Leventhal & Brooks-Gunn, 2000; Sampson, Morenoff & Gannon-Rowley, 2002; Sampson, Raudenbush & Earls, 1997).

Previous research has ably described between-group differences, and established some of the ecological and economic correlates of different levels of pro- and anti-sociality. However, much less progress has been made in understanding the proximate mechanisms that produce (or reproduce) the behavioural and attitudinal differences within the individual. Prevalent proximate explanations for between-group differences invoke cultural transmission and social norms (Henrich et al., 2010). Such explanations are compelling, but merely invoking culture and norms is not in itself an explanation of how individuals acquire them. The psychological mechanisms involved need to be identified (Chudek & Henrich, 2011). Acquisition of local attitudinal patterns might involve lengthy socialization through childhood, followed by relative intra-individual stability, or attitudes could be updated dynamically throughout life according to current context. Explicit verbal instruction might be required. Alternatively or additionally, psychological mechanisms might respond to particular classes of subtle behavioural or physical cues that have, over evolutionary time, been reliably associated with social environments in which particular social behaviours are adaptive. Correlational studies are in general limited in their potential to be able to address these kinds of issues (see Henrich et al., 2012b; van Hoorn, 2012, for recent discussion).

Recent experimental work suggests that mechanisms for calibrating pro- and anti-social behaviours to the local socio-ecology remain highly plastic in adulthood, and are continuously updated using input from the current environment (O’Brien & Wilson, 2011). Peysakhovich & Rand (2013) showed that high- or low-cooperation behaviour could be readily induced amongst experimental volunteers by pre-exposing them to experience of cooperation or defection by others. The authors suggest that people develop heuristics of social cooperation based on experiences of social interaction from their daily lives. These heuristics can be readily and continuously updated by new experience.

Direct personal interaction with others in an environment may not even be necessary to change social behaviour. In a series of field studies inspired by the ‘broken windows’ theory from criminology, Keizer, Lindenberg & Steg (2008) showed that experimentally introducing signs of social disorder, such as graffiti or littering, into the urban environment had remarkably large effects on the propensity of passers-by to litter, violate local rules, and even steal money. These effects were seen immediately, and crossed domains of behaviour; for example, observing that others had littered a public space increased the probability of stealing. Keizer, Lindenberg & Steg (2008, see also Keizer, Lindenberg & Steg, 2013) suggested that individuals have a psychological goal to behave well in the local social context (that is, to uphold norms that are generally agreed to be desirable for all parties). However, the strength of activation of this goal relative to their other goals depends on factors to do with the context and their state. In particular, they are motivated to uphold prosocial norms at cost to themselves only to the extent that others in the social environment are also motivated to do so. The environment provides cues of the motivation of others locally to uphold prosocial norms, in the form of their behaviour and its crystallized consequences in the landscape. These cues can include both disorder (perceptible consequences of others’ not being motivated to uphold prosocial norms), and also order restoration (perceptible consequences of others expending effort in the service of upholding or restoring a prosocial norm). The results of the experimental interventions imply that people are very sensitive to these cues, and use them to continuously calibrate the strength of their own prosocial goals relative to other motivations.

Fessler and colleagues, using psychological priming paradigms, have suggested more specific mechanisms by which such continuous calibration may operate (Fessler & Holbrook, 2013; Schnall, Roper & Fessler, 2010). In particular, witnessing others upholding prosocial goals produces a specific emotion of elevation, which increases the subject’s own prosocial motivation, whilst witnessing the opposite produces declination, a pessimism about others in general that decreases prosocial motivation. We can speculate that, in real-world environments, the continuous calibration via a diet of cues triggering elevation or declination results in a locally distinctive attitudinal stance towards other people in the environment. In social science, this stance is usually operationalized as trust, measured with a question such as ‘To what extent do you think people in general can be trusted?’ Trust measured in this way varies markedly between populations (Bond et al., 2004; Delhey & Newton, 2005; Knack & Keefer, 1997), is predictive of prosocial behaviours (Balliet & Van Lange, 2013; Gachter, Herrmann & Thoni, 2004), and relates to crime rates rates (Kennedy et al., 1998; Roh & Lee, 2013), and the functioning of social institutions (Knack, 2002). Low trust has several consequences. It can produce paranoia, a related and more extreme attitude involving the appraisal that others are trying to cause personal harm (Mirowsky & Ross, 1983). It directly reduces prosocial behaviour, thus leading to the creation of further environmental cues to which others will respond to by reducing their trust. It also reduces motivation to engage in acts of prosocial punishment or social control (Schroeder, Pepper & Nettle, 2013). Communities in which trust is low lack collective efficacy; that is, the capacity of their members to sanction those whose behaviour is antisocial (Sampson, Raudenbush & Earls, 1997), further exacerbating antisociality. Thus, a culture of low trust and low prosociality can become socially entrenched from small beginnings.

If, as suggested by the work described above, the mechanisms calibrating social attitudes remain highly plastic in adulthood, update rapidly, and respond to specific cues in the immediate environment, then people should assimilate to the culture of a population (in the sense of its locally distinctive social attitudes) very rapidly upon encountering it. We hypothesized that putting people temporarily into the environment inhabited by a population, thereby exposing them to the cues that result from the social behaviours of that population, would have a measurable effect on their social attitudes. This paper reports an experiment in which we attempted to test this hypothesis. The setting for our study was two different neighbourhoods within the city of Newcastle upon Tyne. These neighbourhoods have been the focus of ongoing fieldwork for several years (Nettle, 2012; Nettle, Colléony & Cockerill, 2011; Nettle, Coyne & Colléony, 2012; Schroeder, Pepper & Nettle, 2013). They are within a few kilometres of one another and are similar in many regards (size, population, population density, architectural layout, distance from city centre, approximate ethnic composition), but radically different in terms of socioeconomic fortunes. Whereas one neighbourhood (neighbourhood A) is economically thriving and has largely professional homeowner residents, the other (neighbourhood B) has suffered loss of economic activity, blight and continued uncertainty following the deindustrialisation of Newcastle beginning in the 1970s. Neighbourhood B is now classified by the UK government as within the 1% most deprived areas in England. It sustains a rate of crime that is twice that of neighbourhood A, and a rate of violent crime that is 6 times as high (see Nettle, Colléony & Cockerill, 2011, for more detail). We have previously found marked differences between the two neighbourhoods in terms of residents’ play in Dictator, Theft and Third-Party Punishment economic games, and their likelihood of volunteering for a study or returning a lost letter on the pavement (Nettle, Colléony & Cockerill, 2011; Schroeder, Pepper & Nettle, 2013). There is, effectively, a large cultural difference between the two neighbourhoods in terms of pro- and anti-social behaviours and the attitudes that underlie them.

Our experiment had two parts. In the first part, the resident sample, we used our ongoing survey fieldwork amongst the residents to characterize the social attitudes of the residents of the two neighbourhoods. We did this by asking them questions about trust and paranoia. Trust, as previously mentioned, is widely studied in social research. It is generally held to be a central attitudinal variable relevant to the propensity towards pro-sociality and away from anti-sociality, both at the individual and community level (Balliet & Van Lange, 2013). In particular, it is trust in people in general (henceforth social trust), rather than trust in those one knows well (personal trust) that varies most amongst populations and best predicts prosocial outcomes (Uslaner, 2002). Paranoia is the belief that other people are actively trying to harm the subject. It is closely related, conceptually and empirically, to low trust, and has been previously found to be elevated in deprived socioeconomic groups (Mirowsky & Ross, 1983; Ross, Mirowsky & Pribesh, 2001). Paranoia is also related to persecutory symptoms of psychosis that are elevated in dense urban environments (van Os et al., 2001), and amongst psychotic patients, paranoia can be experimentally exacerbated by a short walk in such an environment (Ellett, Freeman & Garety, 2008). We predicted that social trust would be lower, and paranoia higher, amongst residents of neighbourhood B than neighbourhood A.

The second part of our experiment (the visitor sample) tested our main hypothesis regarding assimilation to the social attitudes of a neighbourhood by brief exposure to it. As described below, we randomly assigned a sample of student volunteers to be transported to one or other of the two neighbourhoods, where they completed an urban walk, under the guise of delivering surveys to the houses of the residents. They too completed measures of social and personal trust, and paranoia. We predicted (1) that there would be an effect of which neighbourhood the volunteer had been sent to on their trust and paranoia scores; and (2) that these differences would mirror the pattern of differences between the residents of the two neighbourhoods. If these predictions were met, we would have effectively induced a temporary version of the difference in social attitudes between the residents of the two neighbourhoods by exposure to the cues to which the residents are exposed.

Methods

Ethics statement

All work reported in this paper was approved by the Faculty of Medical Sciences Research Ethics Committee, Newcastle University.

Data availability

The raw data from residents and visitors are downloadable as Supporting Information.

Study sites

Our research was based in the two neighbourhoods, A and B, within the city of Newcastle upon Tyne, Northeast England, that have been described fully in previous papers (Nettle, 2012; Nettle, Colléony & Cockerill, 2011). For this study, the boundaries of neighbourhood B were enlarged slightly compared to our previous work, due to a desire to avoid repeatedly sampling the same residents in surveys. The area into which the expansion occurred is socially similar to the core of neighbourhood B.

Resident sample

Between July 2012 and June 2013, we used the city’s electoral roll to address questionnaires and accompanying letters to randomly chosen residents of each neighbourhood. These were longer questionnaires that formed part of our ongoing fieldwork and which contained measures that are reported elsewhere (Schroeder, Pepper & Nettle, 2013), as well as the two trust measures used in the current study (see Measures below). Residents returned the questionnaires by post, and received £5 in cash as a participation incentive, which was hand-delivered to their houses. From April to June 2013, we modified the resident questionnaire to contain, as well as the trust measures, a measure of paranoia (see Measures below). Response rates were approximately 24% in neighbourhood A and 17% in neighbourhood B. Respondents’ geographical origin was established by asking for the post-code or city in which they had resided at age 10. The total resident sample reported here consisted of 259 responses for trust only, and a further 65 for paranoia and trust.

Visitor sample

In October and November 2012 and April and May 2013, we recruited 52 student volunteers from Newcastle University to visit the two neighbourhoods and post questionnaires through letterboxes of designated resident addresses. They received £5 or course credit for participation, and were aware that they were taking part in an experiment, though not aware of its exact hypothesis. Volunteers did not reside in either neighbourhood and neither neighbourhood was referred to by name to at any point in the session. Their geographical origin was established by asking for the post-code or city in which they had resided at age 10. On arrival at a rendezvous point on the university campus, participants were randomly assigned to be sent to one neighbourhood or the other. They were then taken in groups of 1–4 in a minibus or taxi, with at least one experimenter, to a drop-off point in the neighbourhood, where they were deposited with a packet of questionnaires, a list of resident addresses and a personalised map. They were instructed to find the addresses on foot and deliver the questionnaires, and then return to the waiting vehicle. Participants in the same vehicle set off from the drop-off separately, and were instructed to return after 45 min even if they had not successfully found all target addresses. The time away from the vehicle was 10–48 min (mean±sd 30.39±11.47; precise times were not recorded for the first 14 participants but were not more than 45 min). On return to the waiting vehicle, participants were asked to write down two open-ended comments about the neighbourhood they had just visited. Their answers were prompted as follows. “We would like to know what you thought of the neighbourhood you have been delivering questionnaires in. Please write about two things that seemed important about the neighbourhood. Please tell us why you chose these things”. They were then handed a questionnaire to fill in, ostensibly as part of a separate study. This questionnaire included the measures of trust and paranoia (see Measures below), and a general measure of mood. After completing the questionnaire, they were debriefed and the vehicle returned them to the rendezvous point.

Measures

Our main outcome measures were identical for the resident and visitor samples. In accordance with much previous trust research, we measured each kind of trust with a single item. For social trust, the question was ‘How much do you trust people you meet for the first time?’, whilst for personal trust it was ‘How much do you trust people you know personally?’ The response scale varied from 1 to 10 in each case. For paranoia, we used the conviction subscale of the paranoia checklist from Freeman et al. (2005). This consists of 18 items and is designed to measure paranoid symptoms in non-clinical samples. Cronbach’s α for the paranoia measure was 0.88 in the resident sample and 0.87 in the visitor sample. Visitors additionally rated their current mood on a 10-point scale. The trust and paranoia measures referred to how participants were in their life in general, and for the visitors, made no reference at all to their immediate acute experience, the neighbourhood they just visited, or how they would hypothetically feel if they lived there. The experience they had just had was not alluded to in the questionnaire.

Analysis strategy

All analysis was carried out in SPSS version 19 with a uniform α-value of 0.05 for statistical significance. We had three outcome variables, personal trust, social trust and paranoia. Where there are multiple dependent variables within the same experiment, it is desirable to use a single MANOVA for statistical inference, rather than several ANOVAs, in order to minimize multiple testing. For the resident data, it was unfortunately not possible to use a single MANOVA, since we had social and personal trust scores for 323 and 324 residents respectively, but paranoia scores for only a subset of 65. We therefore conducted separate ANOVA analyses for each outcome variable. In each case, we first performed an ANOVA with neighbourhood as the sole independent variable (henceforth, the simple model). Subsequently we ran a model containing neighbourhood plus sex, age, and – since being in a local minority is associated with paranoid symptoms (Halpern, 1993) – local origin and the neighbourhood by local origin interaction. In the results section, we refer to this as the adjusted model.

For the visitor data, all three outcome measures were taken from the same set of 52 people, so we were able to use a MANOVA to test for an effect of neighbourhood on the set of three measures. Again, a first simple model contained neighbourhood as the sole predictor, whilst a second model adjusted for age and sex. We could not adjust for local origin, since all but one of our visitor participants grew up outside the Newcastle area.

We coded each of the open-ended comments made by the visitors before completing the questionnaire as a basically positive (+), basically negative (−) or unclassifiable (0) reaction to the neighbourhood environment. We thence gave each participant a reaction score, which varied from −2 (two negative comments) to +2 (two positive comments). To establish whether it was the participant’s reaction to the environment they had walked through that was driving any neighbourhood effects on trust and paranoia, we ran additional MANOVA analyses using reaction score as a dependent variable. Finally, for each variable in each neighbourhood, we tested whether the visitor means differed significantly from the estimated marginal means for the residents from the adjusted model. This was done using one-sample t-tests.

Results

Trust and paranoia amongst residents

In the resident sample, social trust and personal trust were moderately positively correlated (r323 = 0.43, p < 0.01). The correlations of the two trust measures with paranoia, though negative, were not significant (social trust: r65 = −0.06, p = 0.62; personal trust: r64 = −0.22, p = 0.09).

For social trust, there was a significant neighbourhood difference in the simple model (F1,322 = 45.48, p < 0.01; means±se: Neighbourhood A 5.00±0.15, Neighbourhood B 3.53±0.16), with trust approximately 0.7 pooled standard deviations higher in Neighbourhood A than B. The neighbourhood difference remained significant in the adjusted model (F1,308 = 29.41, p < 0.01; estimated marginal means±se: Neighbourhood A 4.95±0.16, Neighbourhood B 3.58±0.20). No other effects approached statistical significance in the adjusted model.

For personal trust, there was a significant neighbourhood effect in the simple model (F1,321 = 13.18, p < 0.01; means±se: Neighbourhood A 8.61±0.09, Neighbourhood B 7.97±0.15). This represents a difference of approximately 0.4 pooled standard deviations, with personal trust higher in neighbourhood A. Again, the neighbourhood difference remained significant in the adjusted model (F1,307 = 9.29, p < 0.01; estimated marginal means±se: Neighbourhood A 8.60±0.13, Neighbourhood B 7.98±0.16). No other effects approached significance in the adjusted model.

For paranoia, there was no significant neighbourhood difference in the simple model (F1,63 = 0.001, p = 0.97; means±se: Neighbourhood A 25.14±1.21, Neighbourhood B 25.21±1.58). However, in the adjusted model, the effect of neighbourhood was significant, with neighbourhood B having higher paranoia once age, sex and local origin are controlled for (F1,56 = 4.46, p = 0.04; estimated marginal means±se: Neighbourhood A 24.77±1.31, Neighbourhood B 30.57±2.38). The neighbourhood difference in marginal means in the adjusted model represents approximately 0.7 pooled standard deviations. None of the other effects in the adjusted model was statistically significant, although there were marginally non-significant trends for effects of sex (F1,56 = 3.81, p = 0.06, males higher, estimated marginal means±se: M 29.68±1.79, F 25.66±1.59) and local origin (F1,56 = 3.64, p = 0.06, non-locals higher, estimated marginal means±se: local 25.12±1.22, non-local 30.22±2.38). Figure 1A summarises the resident neighbourhood differences in the three outcome variables.

Figure 1 Levels of social and personal trust (left axis) and paranoia (right axis) for residents of (A) and visitors to (B) the two neighbourhoods.

Bars represent the marginal means from the model adjusting for age, sex and local origin. Error bars represent one standard error.

Trust and paranoia amongst visitors

In the visitor data, social trust and personal trust were moderately positively correlated with each other (r51 = 0.58, p < 0.01), and showed significant or marginal negative correlations with paranoia (social trust: r51 = −0.30, p = 0.03; personal trust: r51 = −0.27, p = 0.06). Time away from the vehicle was not significantly correlated with any of the trust and paranoia measures (social trust: r37 = −0.02, p = 0.91; personal trust: r37 = 0.29, p = 0.09, paranoia: r38 = −0.10, p = 0.57).

In the simple MANOVA, there was a significant effect of neighbourhood visited (F3,47 = 3.68, p = 0.02, Wilk’s λ = 0.81). The neighbourhood effect was driven by a substantial neighbourhood-visited difference in social trust (means±se: Neighbourhood A 4.73±0.46, Neighbourhood B 3.68±0.37; difference equates to 0.5 pooled standard deviations), with visitors to neighbourhood A having the higher social trust. There was a small neighbourhood difference in personal trust, with the higher mean actually found in visitors to neighbourhood B (means±se: Neighbourhood A 7.62±0.40, Neighbourhood B 7.96±0.27; 0.2 pooled standard deviations). We found a substantial difference in paranoia, with paranoia scores being higher in visitors to Neighbourhood B than in visitors to Neighbourhood A (means±se: Neighbourhood A 26.11±1.04, Neighbourhood B 29.64±1.76; 0.5 pooled standard deviations). It should be noted that none of the outcome variables considered in isolation shows a significant neighbourhood difference on an ANOVA (respectively, F1,49 = 3.16, p = 0.08; F1,49 = 0.50, p = 0.48; F1,50 = 3.08, p = 0.09). Nonetheless, the significance of the MANOVA confirms that the effect of neighbourhood visited on the set of outcomes taken together is statistically significant by conventional criteria.

The adjusted model did not change the significance or magnitude of the neighbourhood-visited effect (F3,45 = 3.55, p = 0.02, Wilk’s λ = 0.81; adjusted marginal means very similar to unadjusted means), and the effects of sex and age were not significant. However, in the visitor sample the age range was limited (18–24) and the sex ratio highly unbalanced (10 male, 42 female), so power to detect age and sex effects was low. Means for social and personal trust were similar between the two sexes (means±se: social trust, M 4.10±0.55, F 4.24±0.35; personal trust, M 8.20±0.47, F 7.68±0.28). Mean paranoia was somewhat higher for the male than female visitor participants, in line with the trend for the residents (means±se: M 30.20±1.50, F 27.24±1.21).

The visitor neighbourhood differences are summarised in Fig. 1B. Visitors to neighbourhoods A and B did not differ in self-rated mood after completing their deliveries (means±se: Neighbourhood A 7.12±0.38, Neighbourhood B 7.16±0.39; t49 = 0.08, p = 0.93).

Visitor reaction scores

The open-ended comments given by the visitors to neighbourhood A were uniformly positive (all participants’ scores 2). The comments of visitors to neighbourhood B were much more variable (mean 0.24, s.d. 1.67, range −2 to 2). The reaction score difference between the neighbourhoods was significant (t24 = 5.29, p < 0.01). In a MANOVA with the trust and paranoia measures as dependent variables and reaction score as the independent, the effect of reaction score was significant (F3,47 = 3.43, p = 0.02, Wilk’s λ = 0.82). When both reaction score and neighbourhood visited were entered in the same MANOVA, the effect of neighbourhood visited was no longer significant (F3,46 = 2.33, p = 0.09, Wilk’s λ = 0.87), though reaction score also missed statistical significance (F3,46 = 2.56, p = 0.07, Wilk’s λ = 0.86).

Relationship of visitor responses to the responses of the local residents

To facilitate the direct comparison of residents and visitors for each of the outcome variables, Fig. 2 replots the data from Fig. 1, but with data from residents of and visitors to each neighbourhood shown directly adjacent. To formally compare residents and visitors, we conducted a series of one-sample t-tests comparing the trust and paranoia levels of visitors to each neighbourhood with the trust and paranoia levels of the residents of that neighbourhood. The results of these are given in Table 1. For social trust and paranoia, the pattern is extremely clear: the visitors to a neighbourhood were not significantly different from the residents of the neighbourhood they visited, but were significantly different from the residents of the other neighbourhood (the one they did not visit). For personal trust, the pattern was different. Visitors to either neighbourhood had significantly lower personal trust than the residents of neighbourhood A, and did not differ significantly from the residents of neighbourhood B.

Figure 2 Comparison of resident and visitor levels of trust and paranoia for neighbourhoods A and B.

Bars represent the marginal means from the model adjusting for age, sex and local origin. Error bars represent one standard error.

Table 1 Results of one-sample t-tests comparing the trust and paranoia of the visitors to each neighbourhood to those of the residents of the two neighbourhoods.

Statistically significant differences are underlined. The resident means are marginal means from the model adjusting for age, sex and local origin.

	Compared to residents’ mean of…	
Visitors to…	Neighbourhood A	Neighbourhood B	
Social trust			
Neighbourhood A	t25 = 0.48, p = 0.64	t25 = 2.53, p = 0.02	
Neighbourhood B	t24 = 3.41, p < 0.01	t24 = 0.27, p = 0.79	
Personal trust			
Neighbourhood A	t25 = 2.46, p = 0.02	t25 = 0.91, p = 0.37	
Neighbourhood B	t24 = 2.34, p = 0.03	t24 = 0.07, p = 0.94	
Paranoia			
Neighbourhood A	t26 = 1.29, p = 0.21	t26 = 4.27, p < 0.01	
Neighbourhood B	t24 = 2.77, p = 0.01	t24 = 0.53, p = 0.60	

Discussion

In the first part of our study, we characterized the social attitudes of our two study neighbourhoods using a survey of residents that included measures of trust and paranoia. In accordance with our expectations from previous literature and known facts concerning the socioeconomic context and crime rates, we found that people living in neighbourhood B trusted significantly less, and were significantly more paranoid, compared to people living in neighbourhood A. The neighbourhood effect was larger for social trust than personal trust, and for paranoia it was only detectable once sex, age and local origin had been adjusted for. For none of the outcome variables were sex, age or local origin themselves significant predictors, though, suggesting that we might be detecting consequences of living in the neighbourhood environment, rather than compositional differences – for example of age or ethnic background – between the two populations.

In the second part of the study, we randomly assigned student volunteers to be transported to one or the other neighbourhood and walk around distributing questionnaires to houses. Our prediction (1) was that there would be significant differences in trust and paranoia according to which neighbourhood the participant had been sent to. This prediction was met, with a significant neighbourhood effect on the set of three outcome variables, albeit that none significantly differed between the neighbourhoods when considered in isolation. Our prediction (2) was that the neighbourhood differences amongst the visitors would mirror those seen amongst the residents. This prediction was supported for social trust and paranoia, where the visitor differences were of the same direction and approximately the same magnitude as the differences found amongst the residents. For these two variables, visitors to a neighbourhood did not differ significantly from the residents of that neighbourhood, but did differ significantly from the residents of the other neighbourhood. Thus, for social trust and paranoia, we had effectively induced the attitudinal difference between people in neighbourhood A and those in neighbourhood B through an urban walk lasting 45 min or less. The prediction was not met for personal trust, which was the variable showing the smallest difference amongst the residents. This is comprehensible in retrospect; we had not manipulated participants’ experience with people they knew well, and so there is no reason that the experimental treatment should have any effect on their trust in those people.

There were no significant differences in general mood between visitors who had been to one neighbourhood and those who had been to the other. However, there were marked differences in their qualitative comments about the neighbourhoods, with the comments uniformly positive in neighbourhood A and more mixed in neighbourhood B. There was some evidence that people’s qualitative appraisal of the environment was a mediator of the neighbourhood difference in trust and paranoia, but the strong multicollinearity between neighbourhood and reaction score made this difficult to demonstrate statistically.

These findings thus suggest, in accordance with the findings of other recent studies (Fessler & Holbrook, 2013; Keizer, Lindenberg & Steg, 2008; Keizer, Lindenberg & Steg, 2013; O’Brien & Wilson, 2011; Peysakhovich & Rand, 2013; Schnall, Roper & Fessler, 2010), that the mechanisms regulating social attitudes (and thence behaviours) are highly plastic in adulthood, and can be influenced by cues from the surrounding environment in real time. We believe these findings to have important implications for three areas of research in particular, research in differential psychology, research on neighbourhood effects, and research on cultural evolution.

Implications for differential psychology

Within differential psychology, there is a long-standing debate about the extent to which psychological characteristics should be seen as trait-like rather than immediately situation-driven (Fleeson, 2004). When social factors are shown to be associated with psychological characteristics, the causal nexus is often assumed to be an irreversible developmental effect (e.g., McCullough et al., 2013). The results of this study suggest, however, that trust and paranoia are subject to immediate contextual influence in adulthood, supporting the general importance of current situational variables in driving social behaviours (Zimbardo, 2007). Thus, to explain associations between social deprivation or environmental harshness and behaviour, we may need to consider not just irreversible developmental effects, but also people’s ongoing ‘diet’ of exposure to particular current contextual cues (Nettle, Coyne & Colléony, 2012). This is the process that Buss & Greiling (1999) refer to as enduring situational evocation. Individuals might be quite stable in their trust and paranoia if measured repeatedly over time, but this could simply mean that their exposure to the triggering cues occurs continually. It does not mean that their trust and paranoia would not change if their environment changed.

A number of other recent studies have reached similar conclusions about plasticity in psychological characteristics related to environmental adversity or unpredictability. Mani et al. (2013) investigated the hypothesis that poverty causes poorer cognitive performance. In an experimental study, they showed that people with lower incomes showed poorer cognitive performance than people with higher incomes only when their financial problems were made salient. When financial problems were not salient, there was no difference between the groups. In a related observational study of poor farmers, Mani et al. showed within individuals that cognitive performance declined when money was scarce, and improved again with the harvest when money became available. Kidd, Palmeri & Aslin (2013) studied a classic ‘delay of gratification’ task where children choose between one marshmallow immediately or two after a delay. Variation in performance on this task has been attributed to trait-like differences in self-control. Kidd et al. showed experimentally that giving children an immediate cue that the experimenter was unreliable caused a large reduction in the time the child was able to wait for gratification. Thus, if children from certain social groups show reduced delay of gratification, this may be because they are chronically exposed to cues of unreliability, rather than because their delay of gratification is fixed.

These studies mean that demonstrating differences between groups of people on some characteristic does not mean that those differences are not plastic within each individual, even if they are shown to be stable over time. Cross-sectional studies that purport to show, for example, that a particular social group has low social trust, only really show that people currently in that environment report low social trust. They do not in themselves justify any inference about what those participants would be like if they migrated elsewhere, their state changed, or their public environment was altered. To be clear, we are not claiming that a person’s long-term developmental and cultural history leave no stably internalized influences on social attitudes. It is likely that they do, and indeed, some of the variability in the responses of our samples may well be explained by such influences. We merely wish to draw attention to the relatively strong effects of current situation, and make the methodological point that cross-sectional surveys cannot be used as evidence about how labile social attitudes are within the individual, or what the psychological mechanisms maintaining those attitudes are.

Implications for neighbourhood effects

Neighbourhood effects – associations between neighbourhood characteristics and individual-level outcomes such as health, wellbeing and prosociality – are widely studied in social science, and there are a vast number of correlational studies suggesting their importance (Aneshensel & Sucoff, 1996; Leventhal & Brooks-Gunn, 2000; Pickett & Pearl, 2001; Sampson, Morenoff & Gannon-Rowley, 2002; Sampson, Raudenbush & Earls, 1997). However, the principal challenge with these studies is demonstrating causality (Sampson, Morenoff & Gannon-Rowley, 2002). That is, it is hard to exclude the possibility that people who at the outset have poor health or antisocial tendencies are differentially likely to end up in certain neighbourhoods, rather than the neighbourhood environment causing poor health or antisocial tendencies. Researchers have appreciated that the experimental method is what is required to demonstrate causality (Sampson, Morenoff & Gannon-Rowley, 2002). The (quasi-) experimental designs typically used involve permanent mobility from one type of environment to another (Katz, Kling & Liebman, 2001; Kling, Liebman & Katz, 2007).

There has been much less consideration of the fact that the changes induced by living in a neighbourhood might become manifest in real time, and so, much easier and briefer experiments can also be of interest. Spending 45 min or less in a neighbourhood knowing that there is a vehicle waiting that will take one away is not of course the same as living there. Nonetheless, the fact that social trust and paranoia were so similar for residents of and visitors to a neighbourhood is striking. If a short visit is sufficient to induce detectably lowered trust and heightened paranoia, then how much more powerful must be the effects of living in the place every day? Trust is related to physical and mental health, crime rates, and other social indicators (De Silva et al., 2005; Kawachi, Kennedy & Glass, 1999; Kawachi et al., 1997; Kennedy et al., 1998), whilst paranoia is a clinical psychiatric construct (Freeman et al., 2005), so the outcomes that were affected by our experiment are important for long-term social and health outcomes. Thus, our results tend to support the view that neighbourhood effects are not only causal, but powerful and very rapidly acting. This means that disorder can spread very fast (Keizer, Lindenberg & Steg, 2008), but it does also imply, hopefully, that some of the negative impacts of an environment might be relatively rapid to reverse if environments can be improved (see Keizer, Lindenberg & Steg, 2013). Thus, apparently stable negative consequences of living in a particular environment might actually be labile, adaptively-patterned responses that could quickly change with appropriate social intervention.

Implications for models of cultural evolution

The social attitudes found in particular populations are generally thought of as culturally transmitted (Henrich et al., 2012a; Henrich et al., 2012b; Henrich et al., 2010; Uslaner, 2002). Cultural transmission has been conceptualized as a Darwinian evolutionary process, with the most important change arising through processes analogous to mutation and natural selection (Mesoudi, Whiten & Laland, 2006, though see Claidière & André, 2012). In simple models of cultural evolution, cultural transmission is modelled as occurring once in each lifetime, presumably through socialization in childhood (Boyd & Richerson, 1985). Thereafter, the individual’s cultural traits are fixed and serve as input to the next generation. This maximizes the analogy with genetic evolution. However, our data and that in the other studies reviewed above suggests greater plasticity and lability than such models allow for: social attitudes are continuously updated in adulthood in response to very recent experience. This means that the dynamics of cultural change will be quite different from those of genetic evolution, with cultural patterns able to bloom and fade rapidly in periods much shorter than a generation (Strimling, Enquist & Eriksson, 2009). Darwinian processes of inheritance and selection are not such an appropriate framework for examining this kind of process. Instead, we need bespoke models of cultural dynamics that are built around the actual psychological processes involved in transmission of social attitudes from one person to another, including their intra-individual plasticity. What is needed is to understand the cultural transmission of social behaviours is an empirically-informed ‘epidemiology of representations’ (Sperber, 1985).

Limitations and future directions

Our study had a number of important limitations that should be noted, and future work should seek to overcome these. Our key comparisons in the visitor sample were between subjects. Because of this, we were not able to determine whether individual visitors to neighbourhood A became more trusting as a result of their visit, visitors to neighbourhood B became less trusting, or both. Our methodology also provides no information about which cues are important in explaining the observed effect. We see it is as a proof of principle that being in an environment induces the social attitudes of that environment. Future work using different methodologies will be needed to isolate which cues or interactions are causally important in producing the effect. For example, Hill, Pollet & Nettle (2013) showed experimental volunteers slideshows of street scenes from neighbourhoods A and B, with police presence either prominent or absent in the slideshows. They found that perceptions of safety and social support were lower for neighbourhood B than A, and police visibility had no effect at all. This implies that the high-visibility policing that is a feature of life in neighbourhood B (Nettle, Colléony & Cockerill, 2011) is not one of the main cues people use to calibrate their social perceptions.

Another limitation of our methodology is that it provides a one-off snapshot of the consequences of being in a neighbourhood. We were not able in this experiment to determine the time course of the effects, or establish what would happen with repeated exposure. Although social trust and paranoia were very similar in residents of and visitors to a neighbourhood, the mechanisms producing the differences in the residents may not be exactly the same ones producing the differences in the visitors (though they could be). For example, cues of disorder are very powerful in driving short-term responses (Keizer, Lindenberg & Steg, 2008; O’Brien & Wilson, 2011), but it has been suggested that in the longer term, personal social relationships become more important (O’Brien & Kauffman, 2013). In our data, residents of neighbourhood B showed relatively lowered personal trust, whereas the personal trust of visitors to neighbourhood B was not lowered by their visit. This suggests long-term consequences of living in a neighbourhood that are more than just the immediate visitor reaction. Thus, future work will need to tease out the ways different influences may become more or less important with repeated exposure.

Our resident samples were not representative of the two communities, since only small minorities responded to our surveys. This is hard to avoid in this kind of research, and its consequences are difficult to infer; we may for example have underestimated the true effect size of the neighbourhood differences, if the least trusting and most paranoid residents of neighbourhood B were least likely to respond. There are also important covariate variables that we lacked. We did not know for example how many participants were substance users or had a diagnosed mental illness, and this could have been relevant to understanding variation in paranoia. As for our visitor sample, here we also lacked the sample size and range of measures to assess factors that might have accounted for variation in the response to the neighbourhood, such as cultural and socioeconomic background, and initial level of trust. The visitor sample also had few males, hampering inference about sex differences in attitudes and responsiveness. However, amongst the residents, the only sex difference of note was a near-significant trend for males to have higher paranoia. This is an expected finding (Lewis, 1985), and the means amongst the visitors suggested the same pattern.

Conclusions

Our resident data revealed striking differences in trust and paranoia between people living in two different neighbourhoods. Had we stopped there, we would have assumed that these differences were stable within the individual, and, to the extent they were caused by the neighbourhood, arose from lengthy residence and socialization in those groups. The fact that groups of visitors who spent less than one hour in the neighbourhoods produced very similar patterns of trust and paranoia suggests that immediate contextual experience is relatively important in modulating social attitudes. This may mean that differences in social attitudes between individuals and between populations might be more labile and more context-dependent than previously thought.

Supplemental Information

Supplemental Information 1 The raw datasets from the study

Click here for additional data file.

We would like to thank Bobbie-Jay Hasselby and Anna Wilson for assistance with data collection, and all of the residents and visitors who took part in our research.

Additional Information and Declarations

Competing Interests

Author Contributions

Human Ethics

The authors declare they have no competing interests.

Daniel Nettle conceived and designed the experiments, performed the experiments, analyzed the data, contributed reagents/materials/analysis tools, wrote the paper.

Gillian V. Pepper and Kari Britt Schroeder conceived and designed the experiments, performed the experiments, contributed reagents/materials/analysis tools, revised the manuscript.

Ruth Jobling performed the experiments, revised the manuscript.

The following information was supplied relating to ethical approvals (i.e., approving body and any reference numbers):

The study was approved by the Faculty of Medical Sciences Ethics Committee, Newcastle University, under approvals 00325 and 00503.

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
