# Peer review of "Being there: a brief visit to a neighbourhood induces the social attitudes of that neighbourhood"

_PeerJ, doi:10.7717/peerj.236_

## Round 0.1 · original submission · Minor Revisions

· Academic Editor

Minor Revisions

Please consider the reactions of the reviewers and what they felt was missing. In particular, it is important to state what mechanisms may be operating and what they time course will be to maintain influence. How do you propose to discriminate and test alternative mechanisms in future work? Generally the article is well-written and the results are clear, but you have not satisfied most of the reviewers or me that you have identified specific alternative mechanisms and that you have ways to test these alternative predictions. That would improve the article considerably. In your response please detail the changes you make, which may allow a decision without further review except by me.

Reviewer 1 ·

Basic reporting

Overall, this paper is well-written and concise. The introduction is relevant and gives enough details of the concepts discussed. The methods, results and discussion are generally appropriate, although clarification of a few details should be provided. The research yielded valuable results.

Experimental design

There is only one sentence in the paper about the response rates, saying that "Response rates were approximately 24% in neighbourhood A and 17% in neighbourhood B.", and no interpretation in the limitation section how the response rates or the participation incentive (£5 in cash) can influence the results.

Validity of the findings

As we have no information about the reliability of measures and the administering, the authors should present the Chrobach-alpha of the conviction subscale of the paranoia checklist. It is also unknown whether any of the participants had recieved treatment for psychiatric disorder, and what level of substance use was in resident sample A and B.

·

Basic reporting

This is all fine.

Experimental design

It passes these requirements.

Validity of the findings

They do not raise any particular worries.

Additional comments

This reviews the manuscript “Being there: A brief visit to a neighbourhood induces the social attitudes of that neighbourhood” submitted by Nettle et al. to PeerJ for consideration for publication. The manuscript presents a two-part study examining the social attitudes of the residents of two neighborhoods, and then comparing them to the social attitudes of naïve undergraduate student participants when experimentally exposed to one of the two neighborhoods. Using survey measures of personal trust, social trust, and paranoia, the authors find that the experimentally manipulated undergraduates exhibited social attitudes similar to those of the residents of the neighborhood to which they were exposed. This is then used to make an argument for the adaptiveness of using indicators for local patterns of behavior and interaction as guidelines for one’s own behavioral strategy. In general, I enjoyed the manuscript and found the methodology clever. Strictly speaking, there is little to critique in regards of methodology and analysis. It is also very well-written. For these reasons I think the manuscript is appropriate for publication. I do have a few conceptual (or interpretive, really) concerns, however, that I think the manuscript should probably address.
The manuscript, in evolutionary parlance, is largely a story of ultimate mechanisms, adaptive patterns of interpretation of and response to environment. As a result the question of how the long- and short-term responses are or are not the same is being left open to interpretation. The authors might want to provide more guidance as to the various alternative hypotheses for this. There are some parts of the discussion when it seems to be suggested that the same ambient or direct experiences are shaping both the long-term social attitudes of residents and the short-term social attitudes of the undergraduate participants. At others, it is admitted that we do not know which cues are actually responsible for either, implicitly leaving the door open to there being two (or more) different cue sets at work. For example, in O’Brien & Kauffman (2012) in American Journal of Community Psychology, we discuss how disorder cues might have a short-term effect, but that residents will leverage their access to more direct social information in shaping their behavioral strategies. David Harding in Living the Drama would also argue that there are cultural scripts that reinforce certain patterns of behavior. These would presumably be absent in the undergraduates.
This would then cast some doubt on the assertions reiterated in the concluding paragraph, or at least the strength with which they are made. It is premature to state that when individuals leave their neighborhoods they will shed the social experiences they have had there. Research on the Moving to Opportunity experiment, as confounded as a mobility study might be, would suggest that this is not at all the case. In the current example there was a treatment effect between visits to the two neighborhoods, but there is obviously variation around these means. How much of that variation can be attributed to previously held social attitudes and beliefs, and, in turn, to the influence that one’s native community has had? The home environment will likely leave its trace, even when environmental effects influence the population average. While the authors may be right from a practical standpoint—give people in disadvantaged places a better context and their social attitudes will improve—their interpretation does not give proper attention to the suite of governing mechanisms that are likely at play here.
Last, the section on cultural evolution and the summary of the argument from the El Mouden et al. paper (which admittedly I have not read) leaves me a bit disappointed. Indeed, human social behavior features exceptional levels of plasticity, such that it might be difficult to trace a social strategy, not to mention a single interaction, back to a particular gene and its heritage. That doesn’t mean, however, that there are no such determinative factors that can be identified. If contextual variables, potentially attributable to culture, are inducing these behaviors, their evolution will be critical to the story. (If the pursuant social behaviors themselves influence the continuation or perpetuation of the responsible contextual variables they will then contribute meaningfully to this evolutionary story, as well.) Being that the study is about the influence of context on behavior this strikes me as more satisfying than simply saying that we do not have the proper tools to examine the evolution of these behavioral patterns. It also provides an opening for the evolution of cultural practices and patterns.
The use of a MANOVA seems appropriate enough given the correlations between the variables in the second study. This should probably be clarified, because otherwise the logic for doing so is not made clear. I wonder if the personal trust variable should be maintained in the central analysis, at least for the experiment, as it seems to have been more or less impervious to the treatments. Alternatively, the authors may want to discuss this in terms of the distinction between long- and short-term effects.

Reviewer 3 ·

Basic reporting

Overall, I found the article very well written. The introduction is comprehensive and addresses all of the major empirical questions answered. The discussion similarly follows very well from the introduction and the analyses. The article reports measures clearly and concisely. Some minor typos are present in the manuscript.

Experimental design

I found the experimental design to be of high quality. The comparison of scores from residents of neighborhoods to visitors (who were randomly assigned to neighborhoods) was compelling.

It would be useful for the authors to address individual differences in propensity to trust as a variable of interest in the discussion. Although the random assignment experimental design theoretically mitigates against this problem, individual differences may be valuable to measure in future research.

In general, I would have preferred more commentary on future directions -- e.g., measuring individual differences in trust (e.g., distinguishing between "trait" and "state" variables); using a within-subjects design to measure trust before and after visits to a neighborhood; looking at whether there are any effects of congruency of the types of environments students grew up in and the ones they visited; examining how long the change in trust lasted (was it 20 minutes? an hour? was it only context-specific, and returned to baseline after students left the neighborhood?)

Validity of the findings

The visitor sample size for the experimental portion of the study was small and likely underpowered; however, given the logistic difficulties of performing research of this design, it is understandable and provides results that at the very least are suggestive.

The male-skewed sex ratio of experimental participants was also of concern. Would the authors predict, a priori, that sex would not be relevant? There might be baseline differences in trust between males and females on average, and it is possible that there might be a sex difference in susceptibility to social influences on trust. Perhaps the authors could review the relevant literature in the discussion (this is not an area of direct expertise for myself).

It would be useful if the authors provided descriptive statistics on sex and age differences, even if the results were underpowered (they may be suggestive).

I would have preferred to see exact p-values provided throughout the manuscript.

Why do the authors not present Wilk's lambda statistics for the "Trust and Paranoia among Residents" section?

Was there any relationship between time away from the vehicle and social trust/paranoia measures?

Additional comments

Overall, I found the manuscript to be well-written, well-researched, and of great significance, and I think it is an excellent addition to the literatures on differential psychology, neighborhood effects, and cultural evolution.

·

Basic reporting

First of all, let me thank you for the opportunity to read and review this article. The article is based in a very interesting project that has produced a number of interesting publications. I have only minor comments that are detailed below.

Also, I found a bit surprising that the authors did not refer to the work of Zimbardo on "the power of the situation" or the "Lucifer Effect", which I found strongly related to their study.

The authors also state the following:
"In particular, witnessing others upholding prosocial goals produces a specific emotion of elevation, which increases the subject’s own prosocial motivation, whilst witnessing the opposite produces declination, a pessimism about others in general that decreases prosocial motivation. We can speculate that, in real-world environments, the continuous calibration via a diet of cues triggering elevation or declination results in a locally distinctive attitudinal stance towards other people in the environment." (page 4, § 2, lines 61-66).
I do agree with this observation regarding the emotion of elevation. However, it is also important to point out that elevation does no promote pro-social behaviour (see Haidt, 2006).

Experimental design

I think the design is good. The researchers have first addressed the level of trust and paranoia between neighbourhood A and B. The second part of the study was an experiment in which students were randomly assigned to walk through neighbourhood A or B.

Validity of the findings

I think the findings are valid and actually in line with the work of Zimbardo. Nevertheless, although I do agree on the influence of the environment to our attitudes, I also agree on that we are not a "tabula rasa". The work of Steven Pinker need also to be reviewed, at least in a limitation section. I recommend the authors to discuss the limitations of their studies. For example, who would not be paranoid in a neighbourhood usually recognised as highly violent?

---

## Round 0.2 · accepted · Accept

· Academic Editor

Accept

Your work is interesting and well-described. I wish you well in continuing this line of inquiry.